# Evaluating the Response to Cryopreservation of Ovine Fibroblast Spheroids

**DOI:** 10.3390/biology14101381

**Published:** 2025-10-09

**Authors:** Davide Piras, Federico Olia, Chiara Cosseddu, Daniela Bebbere, Sergio Ledda

**Affiliations:** 1Department of Veterinary Medicine, Obstetrics and Gynecology Clinics, University of Sassari, 07100 Sassari, Italy; f.olia9@studenti.uniss.it (F.O.); c.cosseddu@studenti.uniss.it (C.C.); dbebbere@uniss.it (D.B.); giodi@uniss.it (S.L.); 2CellDynamics iSRL, 40136 Bologna, Italy

**Keywords:** 3D cell culture, fibroblasts, cryopreservation, spheroid

## Abstract

**Simple Summary:**

Long-term storage of three-dimensional cell cultures is essential for tissue engineering and regenerative medicine, yet freezing can damage these complex structures. We produced two sets of spherical aggregates of sheep skin cells, measuring about 140 and 220 μm in diameter, and preserved them by slow freezing in a solution containing dimethyl sulfoxide to limit ice formation. After thawing we quantified survival, energy metabolism, capacity to attach and spread on a surface, weight, mass density and diameter, and we examined genes that signal stress or cell death. Small aggregates rapidly regained normal metabolism, formed continuous cell layers within twenty-four hours and maintained both physical integrity and balanced gene activity. Large aggregates lost compactness and weight, showed extensive cell death in their centers and produced stable layers in only seventy-five per cent of samples. At the molecular level, cryopreservation upregulated stress-related genes such as *HSPA1A* (*HSP70*) and *HSP90AB1*, while downregulating the anti-apoptotic gene *BCL2*, These findings reveal a clear size threshold beyond which cryopreservation compromises viability, offering a practical guideline for building reliable cell banks that can advance wound-repair research, implant design and veterinary therapies.

**Abstract:**

Cell spheroids are widely studied for their potential applications in tissue engineering and regenerative medicine. The present work investigated the effects of cryopreservation on spheroids derived from ovine fibroblasts, depending on spheroid size (140 or 220 µm). Specifically, it explored how cryopreservation impacted several biological and physical parameters including cell damage, viability, metabolism, adhesion, proliferation, and spheroid mass density, weight, and diameter at three time points after thawing. A Live/Dead assay provided a visual assessment of cell damage, cell viability and metabolic activity were assessed by an Alamar Blue assay, and a replating assay evaluated cell adhesion and proliferation capabilities. Spheroid mass density, weight, and diameter were quantified by the W8 Biophysical Analyzer, creating accurate biophysical profiles. Real-time PCR (RT-PCR) analysis was employed to uncover gene expression changes following cryopreservation. Our findings indicate that spheroids measuring 140 µm in diameter largely maintained their biophysical features and cell viability post-cryopreservation, whereas those at 220 µm exhibited a decline in both vitality and mass density. The reduced vitality of 220 µm spheroids likely reflects size-related limitations in cryoprotectant diffusion and stress within the core. Overall, this study provides a comprehensive understanding of how cryopreservation affects ovine fibroblast spheroid biophysics and cellular integrity, laying the groundwork for improved preservation techniques for cell spheroids.

## 1. Introduction

Two-dimensional (2D) cell cultures have historically provided a cornerstone for in vitro studies, greatly enhancing our understanding of cellular behaviors and responses to various stimuli [1,2]. Through decades of research, these models have facilitated insights into the dynamics of numerous cell types and their reactions to external factors [3]. The primary benefits of 2D cultures lie in their simplicity, cost-effectiveness, and the straightforward nature of conducting functional assays [1,2]. Yet, despite their widespread use and accessibility, 2D cultures do not replicate the complex 3D interactions cells experience in vivo, such as cell–cell and cell–extracellular matrix communications [1]. Moreover, artificial flat plastic or glass surfaces alter cellular characteristics, leading to changes in morphology, polarity, differentiation, and proliferation, as well as affecting drug metabolism and other vital functions [4,5] (Figure 1).

Biotechnological methodologies have thus shifted toward 3D cell cultures, bridging the gap between traditional 2D systems and living tissues [1]. These models aim to emulate more closely the natural environment of cells within an organism, improving the physiological relevance of in vitro studies. Indeed, 3D cultures have become critical tools across diverse research areas, including oncology and stem cell biology, and can be created using a range of methods—incorporating hydrogels, biological or synthetic scaffolds—to replicate tissue architecture at multiple scales [3,6]. By reorganizing into 3D structures, cells receive more realistic biochemical and biomechanical stimuli that strongly influence their fate and behavior [1].

Cryopreservation is a key technology for long-term preservation of biological materials and is well established for single dispersed cells. Nevertheless, the growing interest in 3D cellular models for testing responses to drugs or other treatments necessitates effective cryopreservation protocols that preserve complex multicellular architectures. Slow freezing and vitrification are among the main strategies used [7]. In slow freezing, a gradual and controlled cooling process is combined with ~10% dimethyl sulfoxide (DMSO) to avoid intracellular ice crystal formation [8,9]. Vitrification, used in reproductive medicine for gametes and embryos, relies on higher concentrations of cryoprotectants to prevent ice crystallization but may pose an increased risk of cytotoxicity [9,10,11].

Recent advancements in cryopreservation techniques for 3D spheroids extend beyond these conventional approaches. Novel methods employ ice nucleation agents—such as polysaccharides, bacterial proteins, and lipids that help regulate ice formation and minimize cellular damage [12]. Polyampholytes, by controlling osmotic pressure, can also inhibit intracellular ice formation and enhance cell survival [13,14]. Moreover, emerging rewarming strategies (e.g., photothermal rewarming) use nanoparticles with high conversion efficiency for rapid, uniform temperature increases that reduce the risks of devitrification [13,14]. Together, these approaches aim to preserve the structure and function of 3D models, benefiting tissue engineering and regenerative medicine.

Among various cell types grown as spheroids, fibroblasts are of particular interest. They are the most common cells of connective tissue, typically spindle-shaped, and actively participate in synthesizing extracellular matrix components, regulating inflammation, and driving processes like wound healing, tissue fibrosis, and scar formation [15,16,17,18,19,20,21,22,23,24]. Human fibroblasts, for instance, can be derived from patient biopsies, offering critical insights into disease pathogenesis and facilitating the creation of induced pluripotent stem cells [25,26]. Consequently, cryopreservation protocols for fibroblast spheroids may require optimized conditions tailored to the cell source, spheroid size, and freezing approach.

The present work investigates how slow freezing influences the viability, metabolic activity, and biophysical parameters of ovine fibroblast spheroids. To gain deeper insight into the cellular response to cryopreservation, we also evaluated the transcript abundance of a panel of genes. These included four heat shock proteins: {HSPB1 (also known as HSP27), HSPA1A (HSP70), HMOX1 (HSP32) and HSP90AB1 [27,28,29]}, of two proteins implicated in apoptosis (BCL2 and BAX; [30,31]), and a protein involved in cell stress response (CIRBP) [32].

To explore how aggregate size modulates cryopreservation outcome, we deliberately generated two discrete spheroid groups ≈ 140 µm and ≈ 220 µm in diameter simply by halving or doubling the seeding density in Corning^®^ Elplasia^®^ microwells. This approach avoids any mechanical manipulation and mirrors standard practice in high-throughput 3-D culture platforms, where spheroid diameter is routinely tuned by cell inoculum alone [33]. For each group, control samples (CTRL) and slow-frozen samples (SF) were generated.

The selected diameters span the diffusion threshold reported for avascular tissues: oxygen and nutrient supply become limiting when the diffusion path exceeds ~200 µm, leading to hypoxia, acidosis and, eventually, a necrotic core [34,35]. At the same time, both sizes lie within the 100–300 µm range most employed in tissue-engineering and drug-screening workflows, ensuring that our findings remain directly translatable to typical regenerative-medicine applications [36]. Accordingly, the 140 µm group represents a sub-threshold construct expected to equilibrate rapidly with cryoprotectant solutions, whereas the 220 µm group intentionally exceeds the critical radius, allowing us to test the hypothesis that even a modest size increase compromises post-thaw recovery.

By examining size-dependent responses to cryopreservation, our study aims to contribute further understanding of the factors that underpin robust, long-term preservation of 3D cellular models. Although slow freezing is often effective for many cell types, its application to larger spheroids is sometimes problematic due to uneven penetration of cryoprotective agents.

## 2. Materials and Methods

### 2.1. Fibroblast Isolation

Primary fibroblast cultures were obtained from adult sheep ear tissue collected at a regular slaughter facility. Briefly, each biopsy was washed in 70% ethanol, followed by two washes in phosphate-buffered saline (PBS) and two washes in Dulbecco’s Modified Eagle’s Medium (DMEM/F-12) containing 1% penicillin–streptomycin–amphotericin B. The biopsies were then cut into ~1 mm fragments using a scalpel and placed in a 60 mm tissue culture Petri dish with 1 mL of fetal bovine serum (FBS). The dish was incubated at 37 °C with 5% CO_2_. After 24 h, the serum was removed and replaced with DMEM/F-12. After 5 days, the tissue fragments were discarded, and the cells that had migrated from the tissue were cultured until reaching 80% confluence. Cells were then detached with 0.25% trypsin and transferred to 75 cm^2^ flasks, maintained up to the sixth passage.

### 2.2. Spheroid Generation

To produce spheroids of different sizes, Corning^®^ Elplasia^®^ Microplates (Corning Inc., Corning, NY, USA) were used.

Prior to cell seeding, the plates (400 microcavities per well) were pre-wetted to eliminate residual air bubbles within the microwells. Each well was filled with 50 µL of complete growth medium (Dulbecco’s Modified Eagle Medium supplemented with 10% fetal bovine serum), followed by centrifugation at 500× *g* for 1 min at room temperature. This pre-wetting step ensured uniform filling of the microcavities and promoted consistent distribution of cells during subsequent seeding.

Two different cell seeding densities were tested:1 × 10^5^ cells per microplate (≈2000 cells per microwell) 140-micron group.1 × 10^6^ cells per microplate (≈4000 cells per microwell) 220-micron group.

Cells were maintained in DMEM/F-12 for up to 6 days, with medium changes every 2 days. On day 7, ~200 spheroids were harvested for cryopreservation via slow freezing, while other spheroids remained in the microplates as the non-frozen controls.

Spheroids were harvested by gentle aspiration with wide-bore pipette tips. To create the wide bore, the distal 2–3 mm of a standard 200 µL tip was removed with a sterile scalpel, reducing shear forces and minimizing contact between the tip walls and the spheroid surface. Using this approach, ≈200 spheroids were withdrawn from each well that originally contained ≈400 spheroids; the remaining spheroids constituted the untreated control group (CTRL).

The following assays were performed on cryopreserved and control spheroids: Alamar Blue (Thermo Fisher Scientific, Waltham, MA, USA) viability assay, replating assay, Live/Dead analysis (Thermo Fisher Scientific), W8 Biophysical Analyzer (CellDynamics i.s.r.l, Bologna, Italy) measurements, and gene expression analysis by real-time PCR.

### 2.3. Cryopreservation by Slow Freezing and Thawing

On day 6 of spheroid growth, spheroids were collected and resuspended in a freezing solution (90% FBS and 10% DMSO) at a concentration of ~200 spheroids mL^−1^ in 1.5 mL cryovials. Slow freezing was carried out using Mr Frosty™ (Thermo Fisher Scientific), which cools samples at ~1 °C min^−1^ in a −80 °C freezer overnight; the following day (day 7) the vials were transferred to liquid nitrogen (−196 °C) for storage. We kept unfrozen spheroids of both diameters in their original Elplasia^®^ plates, maintained them in a 37 °C, 5% CO_2_ incubator, and renewed the culture medium every two days. While the treated spheroids were undergoing the freezing protocol, control (CTRL) spheroids were processed in three equal batches to generate perfectly time-matched reference points:On day 6, the first third of CTRL spheroids was harvested and assayed, providing the 24 h control data set.On day 7, the second third was collected for the 48-h control endpoint as the frozen vials were moved from −80 °C to liquid nitrogen.On day 8, the final third of CTRL spheroids was harvested for the 72-h control analysis.

After 24 h in liquid nitrogen (day 8), cryovials were removed and placed in a 37 °C water bath under gentle agitation for ~50 s. Thawed spheroids underwent their own 24 h post-thaw assays in parallel with the 72 h CTRL measurements, and were then returned to standard culture conditions (37 °C, 5% CO_2_) to allow collection of the 48 h and 72 h post-thaw data sets, mirroring the timeline applied to the controls. Next, spheroids were resuspended in 8 mL of DMEM/F-12 within a Petri dish coated with 2.5% agarose (to prevent adhesion) and incubated at 37 °C with 5% CO_2_.

### 2.4. Alamar Blue Assay

Spheroid viability was evaluated by Alamar Blue. Briefly, 10 spheroids per group, were transferred (in triplicate) into a 10% Alamar Blue/DMEM (phenol-red-free) solution in a 96-well plate and incubated at 37 °C with 5% CO_2_ for 4 h. Absorbance was read at 570 and 600 nm on a Spectrostar Nano (BMG Labtech, Ortenberg, Germany), and the percentage of Alamar Blue reduction was calculated. Measurements were performed at 24, 48, and 72 h post-thawing.

To calculate the % Reduction of Alamar Blue Reagent using absorbance readings:(1)$$\%Reduction\;of\;Alamar\;Blue\;Reagent=\frac{\left(Eoxi600\;\times \;A570\right)-\left(Eoxi570\;\times \;A600\right)}{\left(Ered570\;\times \;C600\right)-(Ered600\;\times \;C570)}\;\times 100$$
where: *Eoxi570* = molar extinction coefficient (E) of oxidized Alamar Blue Reagent at 570 nm = 80,586;

*Eoxi600* = E of oxidized Alamar Blue Reagent at 600 nm = 117,216;

*A570* = absorbance of test wells at 570 nm;

*A600* = absorbance of test wells at 600 nm;

*Ered570* = E of reduced Alamar Blue at 570 nm = 155,677;

*Ered600* = E of reduced Alamar Blue at 600 nm = 14,652;

*C570* = absorbance of negative control well (media, Alamar Blue Reagent, no cells) at 570 nm;

*C600* = absorbance of negative control well (media, Alamar Blue Reagent, no cells) at 600 nm;

Alamar Blue readings were expressed as mean ± SD (*n* = 3 independent wells per condition). Differences between CTRL and SF at each time point were analyzed with a two-tailed, unpaired Student’s *t*-test. Prior to testing, residuals were checked for normality (Shapiro–Wilk).

The significance threshold was set at *p* < 0.05. All statistics were performed in GraphPad Prism 8.0; significant differences are denoted in the figures by asterisks (* *p* < 0.05, ** *p* < 0.01, *** *p <* 0.001).

### 2.5. Replating Assay

Within 15 min of thawing, we dispensed ten spheroids into a single well of a tissue-culture-treated 96-well plate. This set-up was replicated in three independent wells per condition, yielding 30 spheroids for the slow-frozen group (SF) and, in parallel, 30 spheroids for the matched unfrozen controls (CTRL). Plates were then returned to 37 °C and 5% CO_2_ and observed for the next 48 h.

Cell attachment and early outgrowth were quantified by bright-field microscopy (Nikon Eclipse Ts2, Nikon Corporation, Tokyo, Japan):Time points: 8, 24 and 48 h post-seeding.Scoring criteria: a spheroid was counted as “re-plated” when a radial monolayer had spread ≥ 1 spheroid diameter beyond the aggregate edge.

Using this method, CTRL and cryopreserved 140 µm groups reached 100 % re-plating by 48 h, whereas the cryopreserved 220 µm group plateaued at ~75 %.

### 2.6. Live/Dead Staining

The viability of spheroids was also assessed 24 h after thawing using calcein-AM, propidium iodide (PI), and Hoechst. Briefly, 10 spheroids from three groups (cryopreserved and control) were incubated with 1 µL of calcein-AM (10 µg/µL; Invitrogen, Waltham, MA, USA), 1 µL of PI (10 µg/µL; Sigma-Aldrich, St. Louis, MO, USA), and 1 µL of Hoechst (10 µg/µL; Thermo Fisher Scientific) for 30 min at 37 °C with 5% CO_2_. After incubation, spheroids were transferred onto a microscope slide and viewed under a fluorescence microscope (Leica CTR 6000, Leica Microsystems, Wetzlar, Germany).

### 2.7. W8 Biophysical Analyzer

For biophysical measurements, 20 cryopreserved and 20 control spheroids for each group (140 µm, 220 µm) were transferred to 15 mL Falcon tubes containing 7 mL of W8 analysis solution (CellDynamics, isrl, Bologna, Italy) and analyzed with a W8 Physical Cytometer (CellDynamics) at 24, 48, and 72 h post-thaw, according to the manufacturer’s instructions. The W8 Physical Cytometer measures size, weight, and mass density of spheroids in sterile conditions.

The W8 is a flow-based “physical cytometer” that measures diameter, weight and mass density on single spheroids as they free-fall through a micro-fluidic analysis channel filled with a liquid of known viscosity (η) and density (ρₗ). Contact-free free-fall: A peristaltic pump pulls one spheroid at a time from a Falcon tube into a micro-channel, then stops the flow. The spheroid falls through still fluid, satisfying Stokes’ law, so its terminal velocity gives the weight.

Automatic diameter capture: While the aggregate rotates during the 2–3 mm fall, a camera records a burst of brightfield images. The software extracts Feret diameters from every frame and averages them, delivering a sub-micron diameter that is more reliable than a single microscope snapshot.

Mass density: The software combines weight and volume (π d^3^/6) to output the mass density.

Shear stress is kept to minimum, the peristaltic pump feeds each spheroid slowly and centrally, so the aggregate never contacts the channel walls; once inside the measuring zone, the flow is stopped, eliminating surface shear altogether; and because the circuit is closed and sterile, the spheroids remain viable for any downstream assay.

### 2.8. Gene Expression Analysis

#### 2.8.1. RNA Isolation and Reverse Transcription

All samples (~20 spheroids each) were previously stored at −20 °C in 20 µL of RNAprotect^®^ Cell Reagent (Qiagen, Hilden, Germany). Total RNA was isolated using the RNeasy Micro Kit (Qiagen, Hilden, Germany), following the manufacturer’s instructions. Isolated RNA was eluted in 15 µL of RNase-free water and immediately reverse transcribed in a 20 µL reaction containing 75 mM KCl, 50 mM Tris–HCl (pH 8.3), 5 mM dithiothreitol (DTT), 3 mM MgCl_2_, 1 mM dNTPs, 2.5 µM random hexamer primers, 20 U RNase OUT, and 100 U SuperScript III Reverse Transcriptase (Invitrogen, Carlsbad, CA, USA). The reaction tubes were incubated at 25 °C for 10 min, followed by 42 °C for 1 h, and finally 95 °C for 5 min to inactivate the enzyme.

#### 2.8.2. Real-Time Polymerase Chain Reaction

Relative quantification of transcripts was performed by Real-Time Polymerase Chain Reaction (Real-Time PCR) in a Rotor-Gene QMDx 5plex HRM system (Qiagen). The amplification was performed in a 15 µL reaction volume containing:7.5 µL of 2× Quantinova SYBR Green PCR Master Mix (Qiagen);200 nM of each primer (forward and reverse; Table 1);1 µL of cDNA equivalent to ~1 spheroids.

The PCR protocol consisted in two initial incubation steps (50 °C for 5 min and 95 °C for 2 min), followed by 40 cycles of amplification (95 °C for 15 s and a gene-specific annealing temperature—see Table 1—for 30 s). A melting curve program (65–95 °C, starting fluorescence acquisition at 65 °C, and measuring at 10 s intervals until temperature reached 95 °C), and finally a cooling step to 4 °C. Fluorescence data were acquired during the annealing steps.

To minimize handling variability, all samples were analyzed in the same run using a PCR master mix containing all components except the cDNA. PCR products were analyzed by generating a melting curve to check the specificity and identity of the amplification product. For each primer pair, the efficiency of PCR reaction was previously assessed by building a standard curve with serial dilutions of a known amount of template, covering at least three orders of magnitude, so that the calibration curve’s linear interval included the interval above and below the abundance of the targets. Only primers achieving an efficiency of reaction between 90 and 110% (3.6 > slope > 3.1) and a coefficient of determination r2 > 0.99 were used for the analysis.

Normalized expression levels were calculated against the endogenous reference genes *YWHAZ* (tyrosine 3-monooxygenase), and *SDHA* (succinate dehydrogenase). Real-time RT-PCR data are presented as ΔCq ± SEM.

### 2.9. Statistical Analysis

All statistical analyses were performed using GraphPad Prism version 8.0 (GraphPad Software, San Diego, CA, USA). Data are reported as mean ± standard deviation (SD) unless otherwise indicated. Normality was tested using the Shapiro–Wilk test, and variance homogeneity was assessed with the Levene test.

Alamar Blue^®^ assay and W8 Analyzer measurements: differences between CTRL and SF spheroids at each time point (24, 48, 72 h) were analyzed by two-tailed, unpaired Student’s *t*-test. Outliers were excluded when K > 1.5.Gene expression data: after verification of distribution (Kolmogorov–Smirnov test), comparisons were performed with a General Linear Model ANOVA.Significance threshold: *p* < 0.05.Figure notation: statistical significance is indicated by * (*p* < 0.05), ** (*p* < 0.01), *** (*p* < 0.001).Replicates: For each experimental condition, three independent biological replicates were included, with multiple spheroids analyzed per replicate as specified in the Materials and Methods subsections.

## 3. Results

### 3.1. Replating

At 8 h post-seeding, initial adhesion was visible in all CTRL spheroids but in none of the cryopreserved 220 µm group (SF-220). By 24 h, every CTRL spheroid and every cryopreserved 140 µm spheroid (SF-140) had formed a continuous monolayer, whereas 75% ± 8% of the SF-220 spheroids had done so (*n* = 30 per group, *p* < 0.05 vs. CTRL-220). After 48 h the attached spheroids displayed further radial expansion; in several wells, adjacent monolayers from CTRL and SF-140 cultures had merged, suggesting continued proliferation and lateral migration. No additional SF-220 spheroids are attached after the 24 h mark. These data indicate that cryopreservation delays—but does not completely prevent adhesion and outgrowth, with the effect confined to the larger (220 µm) aggregates (Figure 2).

### 3.2. Alamar Blue

The Alamar Blue assay confirmed a size-dependent loss of metabolic activity (Figure 3). In the 220 µm group, resazurin reduction fell by 20.9% at 24 h, 17.3% at 48 h and 17.9% at 72 h versus their unfrozen controls (** *p* < 0.001 for all). By contrast, the 140 µm spheroids showed only minor drops 2.4% at 24 h, 4.2% at 48 h and 5.8% at 72 h with statistical significance reached solely at the 48 h point (*p* < 0.05). These findings indicate that larger spheroids experience a sharper and more persistent reduction in metabolic activity after cryopreservation, consistent with a greater vulnerability to ice-related damage and cryoprotectant stress.

(A) 140 µm spheroids: a modest decline becomes significant only at 48 h (*p* < 0.05).(B) 220 µm spheroids: activity is significantly lower at every time point (** *p* < 0.001). Scale bar not applicable.

### 3.3. W8 Biophysical Analysis

The W8 analysis revealed that 140 µm spheroids (Figure 4) preserved their biophysical profile: mass density deviated from unfrozen controls by only +4.1% at 24 h, −3.3% at 48 h and +2.5% at 72 h, and the brief change in weight and diameter detected at 48 h disappeared by 72 h (Figure 5).

Conversely, 220 µm spheroids showed sustained alterations. Their mass density dropped by −25.8% at 24 h, −30.2% at 48 h and −29.5% at 72 h, with comparable losses in weight and diameter (Figure 5, ** *p* < 0.001). These data indicate that small aggregates quickly return to baseline after cryopreservation, whereas large aggregates retain a long-lasting reduction in density and size that mirrors their lower metabolic activity.

#### 3.3.1. Spheroids of 140 μm

The biophysical parameters of 140 µm spheroids were first analyzed. Mass density, weight, and diameter were measured at 24, 48, and 72 h in both control and slow-frozen groups. The results are summarized in Figure 4, while detailed numerical values are reported in Table 2.

#### 3.3.2. Spheroids of 220 μm

The biophysical parameters of 220 µm spheroids were analyzed at 24, 48, and 72 h. Mass density, weight, and diameter were measured in both control and slow-frozen groups, allowing the comparison of temporal changes. The results are summarized in Figure 5, while detailed numerical values are reported in Table 3.

## 4. Live Dead

### 4.1. Live/Dead Imaging

Qualitative Live/Dead imaging at 24 h supported the biochemical findings (Figure 6). CTRL spheroids of both sizes contained only occasional PI-positive nuclei, evenly distributed across the section. Cryopreserved 140 µm spheroids were similar, showing no obvious necrotic focus. In contrast, cryopreserved 220 µm spheroids exhibited an intense PI signal confined to the interior, consistent with a necrotic core and impaired cryoprotectant penetration. This size-dependent pattern reinforces the conclusion that larger aggregates are more susceptible to spatially localized cryoinjury.

### 4.2. Gene Expression Analysis

The expression of all analyzed genes was confirmed in the spheroids.

Preliminary analysis of the four potential reference genes *ACTB*, *RPL19*, *SDHA* and *YWHAZ* showed significant differences in the expression pattern of *ACTB* and *RPL19* (*p* < 0.05) between control group and cryopreserved samples. Consequently, only *SDHA* and *YWHAZ* were used as reference genes for target gene normalization.

Expression analysis of specific transcripts considered the variables “treatment” (cryopreservation or not) and “time of in vitro culture post-cryopreservation” (0 h, 24 h, 48 h or 72 h).

Cryopreservation caused an increase in *HSP90AB1* and *HSPA1A* transcript abundance in the cryopreserved-thawed spheroids compared to the control group (*p* < 0.05; Figure 7). On the contrary, *BCL2* showed transcript downregulation in the cryopreserved group compared to the control group (*p* < 0.05; Figure 7).

After normalization, *RPL19* showed lower levels of mRNA in the cryopreserved-thawed spheroids (*p* < 0.05),while *ACTB* did not show any significant difference between the control group and the cryopreserved group (Figure 7).

Expression of *HSPB1*, *HMOX1*, *BAX* and *CIRBP* was not affected by cryopreservation (Figure 7).

No genes showed significant differences in relation to the duration of in vitro culture or by the interaction between time and treatment.

## 5. Discussion

Cryopreservation is a widely used technique for preserving biological materials, including cells, tissues, and organs, by freezing them at extremely low temperatures. Often, cell cryopreservation includes cells derived from 2D culture systems, while only in recent years there has been growing interest in using cryopreservation to preserve spheroids.

Our study investigated the feasibility of cryopreservation of two groups (140 and 220 µm) of ovine fibroblast spheroids generated from different cell numbers (2000 or 4000), using the slow freezing technique and 10% dimethyl sulfoxide (DMSO) as cryoprotectant. The results indicated that the metabolic activities of the two groups of spheroids showed significant differences. The 220 µm spheroids exhibited a decrease in the ability to reduce resazurin to resorufin (Alamar Blue assay, Figure 3) compared to the controls at all measured time points (24, 48, 72 h), along with a significant decrease in mass density at all time points compared to the controls. Meanwhile, the 140 µm spheroids showed a significant decrease in metabolic activities only at 48 h.

A likely reason for the temporary increase in size and apparent weight of cryopreserved spheroids at 48 h is a short-lived swelling phase that follows thawing. In the first day after freezing, some cells experience osmotic stress and mild membrane damage, take up extra water, and lose tight packing. This collective swelling makes the whole aggregate look larger and heavier, even though overall cell health is momentarily reduced. As the most damaged cells die or recover and the surviving cells regain normal volume regulation, the spheroid gradually compacts again, so by 72 h both diameter and weight return toward baseline (Figure 4 and Figure 5). The observed difference in viability between the two spheroid groups might be due to the higher initial mass density of the 220 µm spheroids compared to the 140 µm group (1029 fg µm^−3^ vs. 1022 fg µm^−3^). This increased density could lead to greater difficulty in cryoprotectant penetration into the inner layers of the 220 µm spheroids, due to their strong cell-to-cell junctions and 3D structures, resulting in increased ice crystal formation capable of breaking cell membranes.

Additionally, it was observed that both the 140 µm and 220 µm cryopreserved spheroids did not show monolayer formation during the first 8 h after replating, a behavior observed in the controls. However, at 24 h, 100% of the 140 µm cryopreserved spheroids had formed monolayers, while only 75% of the 220 µm cryopreserved spheroids formed monolayers, confirming significant cell damage due to cryopreservation. It was also observed that the edges of the cryopreserved spheroids are less uniform, showing a loss of cellular material compared to the controls (Figure 2).

The difference in viability compared to the controls is also visible through images acquired using fluorescence with the LIVE/DEAD assay (Figure 6). There is a more pronounced and widespread presence of dead cells in the cryopreserved 220 µm spheroids compared to the controls. Dead cells are also observed in the cryopreserved 140 µm spheroids but to a lesser extent than in the controls. To better understand the molecular pathway potentially solicited in larger spheroids, that display reduced viability, metabolic activity post-thawing and biophysical perturbations, we investigated the transcript abundance of genes implicated in cell stress response or apoptosis.

The molecular analysis revealed significant alterations in the expression of stress-related and apoptotic genes following cryopreservation. Notably, *HSPA1A* (*HSP70*) and *HSP90AB1* were upregulated in cryopreserved spheroids, consistent with previous studies indicating that cryopreservation induces cellular stress and heat shock protein activation to mitigate damage [27,37]. The overexpression of *HSP70* has been associated with increased cell survival post-thawing, as it facilitates protein refolding and prevents aggregation [38]. In contrast, *BCL2*, an anti-apoptotic gene, exhibited significantly higher expression in control spheroids, suggesting reduced apoptotic resistance in cryopreserved samples (*p* < 0.05). This aligns with findings by Kopeika et al. [28], where decreased *BCL2* levels correlated with enhanced apoptotic activity post-cryopreservation (Figure 7) [30,31].

Interestingly, the downregulation of *RPL19* in cryopreserved-thawed spheroids (*p* < 0.05) points to its potential role in maintaining cellular homeostasis, as ribosomal proteins are essential for protein synthesis and cellular repair mechanisms [28]. Conversely, the lack of significant changes in *HSPB1*, *HMOX1*, *BAX*, and *CIRBP* suggests that their roles may not be directly influenced by cryopreservation or that their response is temporally delayed. These findings highlight the complex interplay between cellular stress responses and apoptosis in 3D spheroid models and underscore the importance of optimizing cryopreservation protocols to minimize molecular perturbations.

The result of this study confirms that cryopreservation of ovine fibroblast spheroids by slow freezing can lead to a survival rate above 75% [11] for individual samples, allowing the generation of biobanks of samples. However, the study aimed to determine potential differences in terms of cellular damage among cryopreserved samples with varying cellular densities, as well as to measure their biophysical characteristics for the first time and understand how these might affect cryopreservation. It was found that larger samples (220 µm in diameter) with a higher mass density suffered more from slow freezing. This hypothesis could be validated by generating samples with increasing mass density and subjecting them to slow freezing.

It is well known that cryopreservation of multicellular structures or cell multilayers is more complicated than for single cells due to reduced permeability to cryoprotectants and the potential release of these agents after the thawing process. Some of these challenges may be affecting our experiments, where we observed that viability and functional activity were better in 140 µm samples compared to the 220 µm samples.

The response of 3D cellular spheroids to cryopreservation could also be influenced by the freezing technique employed. It has been observed that vitrification may be superior to slow freezing [39]. However, there is still debate regarding which method is most efficient, as cell viability alone may not be sufficient to determine optimal cell functionality. Estimations of cell functional performance must also be considered. Improvement in cell metabolic functions has also been achieved by adding non-penetrating cryoprotectants, such as sucrose [39]. Several studies have previously addressed the cryopreservation of multicellular spheroids using different approaches. Arai et al. [11] reported successful cryopreservation of scaffold-free tubular constructs assembled from spheroids, showing that slow freezing with DMSO can preserve structural integrity within engineered tissues. Truong et al. [40] emphasized the technological potential of “ready-to-use” cryopreserved spheroids and outlined critical process parameters that influence post-thaw viability. Bissoyi et al. [41] demonstrated that supplementing conventional DMSO-based protocols with polyampholyte cryoprotectants improved the survival of hepatocyte spheroids, highlighting the role of macromolecular agents in enhancing protection. More recently, Ishizaki et al. [42] showed that zwitterionic cryoprotectants can further increase post-thaw recovery when combined with DMSO, underscoring the benefits of advanced cryoprotective formulations.

In comparison, our study introduces a complementary perspective by combining metabolic, viability, and gene expression analyses with biophysical profiling at single-spheroid resolution. Specifically, we identified a clear size-dependent vulnerability, as 220 µm spheroids exhibited reduced metabolic activity, loss of mass density, and stress-induced molecular alterations, whereas 140 µm spheroids largely retained their structural and functional integrity. This finding brackets the critical ~200 µm diffusion threshold and suggests that size per se represents a major determinant of cryopreservation outcome, in line with known limitations of cryoprotectant penetration and oxygen/nutrient gradients. Taking together, our results expand the existing literature by providing quantitative evidence for a size boundary in fibroblast spheroids under standard slow-freezing conditions, while also pointing to the potential of integrating advanced cryoprotectants as described in previous studies to overcome these limitations.

## 6. Conclusions

Cryopreservation of 3D fibroblast spheroids is a valuable technique that enables the long-term storage and utilization of these cellular models in various biomedical applications. Optimization of cryopreservation protocols is critical to maximizing cell viability and maintaining the physiological relevance of the spheroids. The data presented in this study highlights the importance of tailoring cryopreservation methods to the specific characteristics of spheroids, such as size and cellular density, as these factors significantly influence post-thaw viability and functionality.

The observed differences in metabolic activity, mass density, and gene expression profiles between 140- and 220-micron spheroids suggest that larger spheroids are more susceptible to cryopreservation-induced damage. This vulnerability may stem from the limited penetration of cryoprotectants into the core of larger spheroids, resulting in heterogeneous freezing and increased ice crystal formation. Future studies should explore alternative cryopreservation techniques, such as vitrification, to determine whether faster cooling rates can mitigate intracellular ice formation and improve the survival of larger spheroids.

Where the exact “safe size” sits are likely to vary depending on the specific cell type. Spheroids that secrete dense extracellular matrix, such as chondrocytes or osteogenic MSCs, tend to restrict diffusion and will likely need to stay below ~150 µm in diameter, unless extra steps improve cryoprotectant uptake [43,44]. Highly metabolic cells, such as cardiomyocytes or pancreatic β-cells, reach oxygen limits more rapidly and therefore benefit from smaller diameters as well [45]. In contrast, looser and more porous assemblies, like hepatocyte or endothelial spheroids, may tolerate diameters approaching 250 µm, before viability is compromised [46]. Finally, every cell lineage is likely to require its own diameter threshold; however, starting with smaller spheroids or enhancing cryoprotectant penetration is the safest strategy.

Ultimately, the successful cryopreservation of spheroids will enhance the reproducibility and scalability of 3D cell culture systems, facilitating their integration into preclinical and clinical workflows. By addressing the limitations identified in this study, researchers can advance the development of robust cryopreservation strategies that preserve both the structural integrity and biological functionality of spheroids, thereby maximizing their utility in various scientific and medical applications.

## Figures and Tables

**Figure 1 biology-14-01381-f001:**
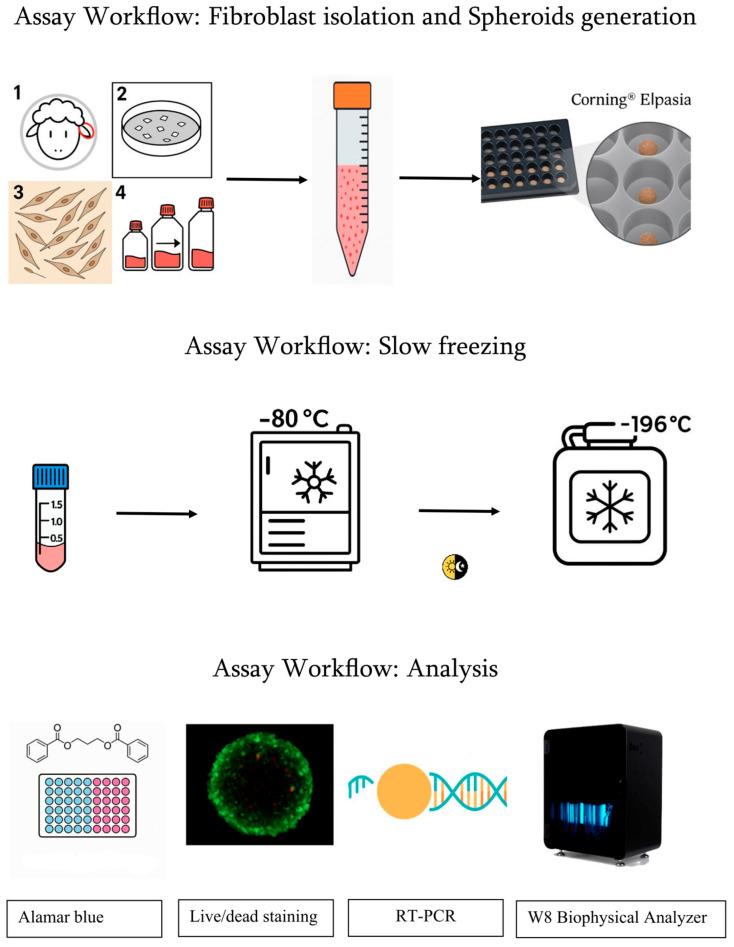
Overview of the experimental workflow, from ovine fibroblast isolation and monolayer expansion to 3D spheroid formation and cryopreservation via slow freezing. Post-thaw, assays were used to evaluate metabolic activity (Alamar Blue), cell viability (Live/Dead), gene expression (RT-PCR), and biophysical properties (W8 Biophysical Analyzer).

**Figure 2 biology-14-01381-f002:**
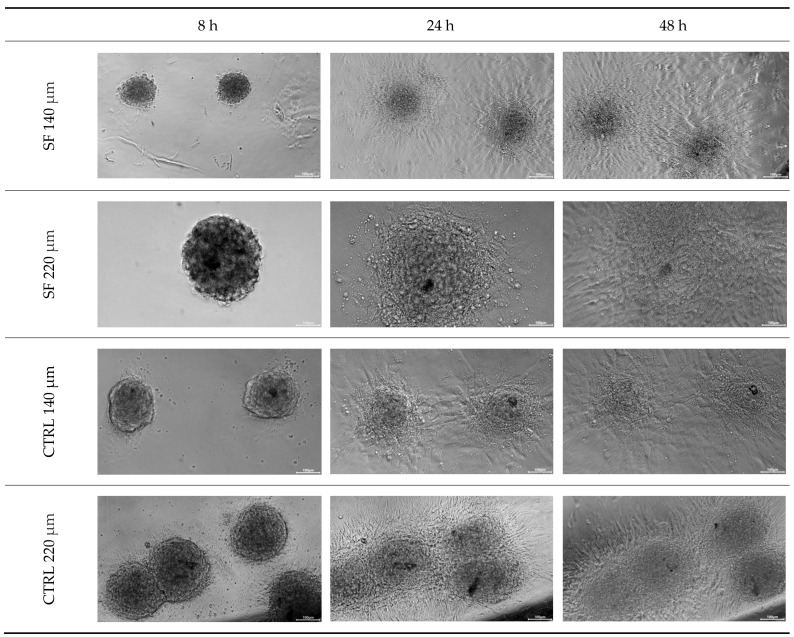
Replating assay. Phase-contrast images of control (CTRL) and slow-frozen (SF) spheroids, 140 µm and 220 µm, taken at 8, 24 and 48 h after seeding. CTRL and SF-140 spheroids attach and spread into a confluent monolayer within 24 h (100% success), whereas only 75% of SF-220 spheroids reach the same stage. By 48 h the attached spheroids show radial outgrowth and occasional fusion of neighboring monolayers. Scale bar = 200 µm. Images were acquired with a Nikon Eclipse Ts2 inverted microscope.

**Figure 3 biology-14-01381-f003:**
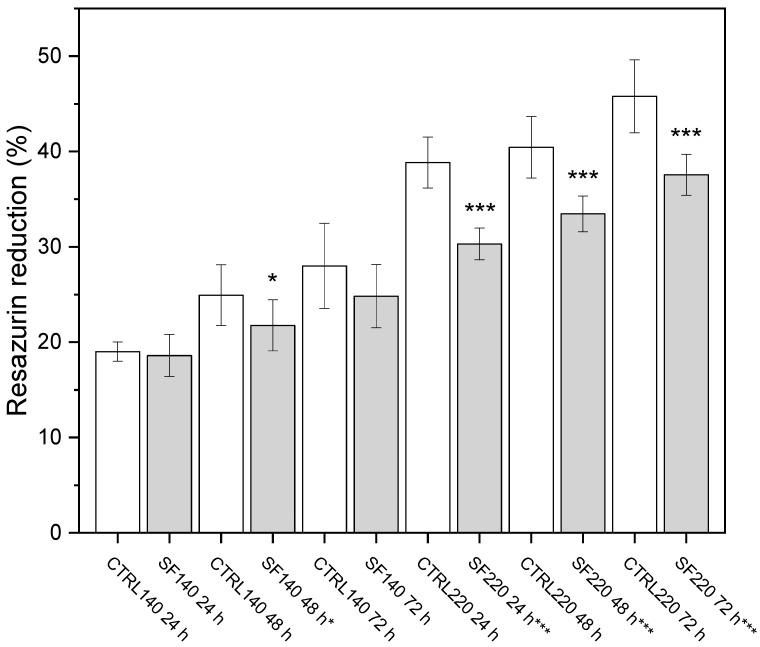
Metabolic activity after thawing. Alamar Blue reduction (mean ± SD, *n* = 3) for control (CTRL, white bars) and cryopreserved (SF, gray bars) spheroids at 24, 48 and 72 h. *p*-values < * 0.05; *p*-values < *** 0.001.

**Figure 4 biology-14-01381-f004:**
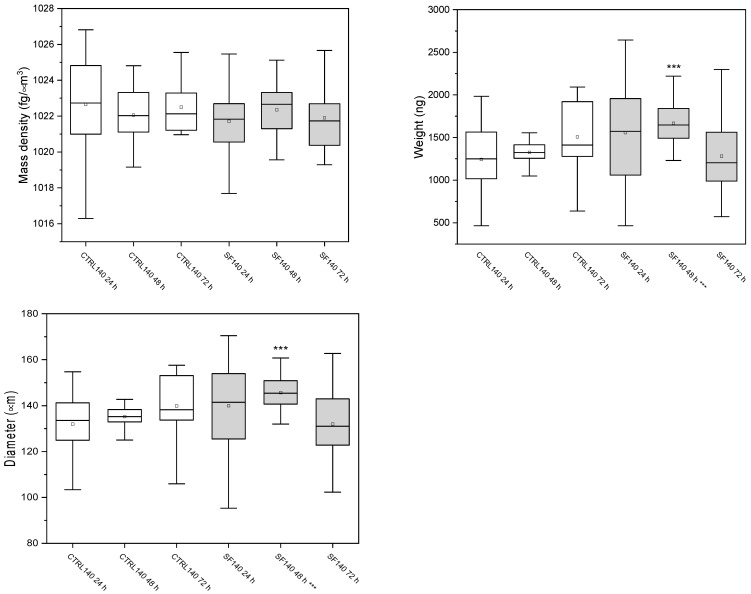
Box plot representation of mass density, weight and diameter of control (CTRL, white bars) and slow-frozen (SF, gray bars) spheroids of 140 µm measured at 24, 48 and 72 h. The line inside each box indicates the median; the lower and upper box boundaries correspond to the first and third quartiles (Q1 and Q3); the whiskers extend to 1.5 × IQR. In 140 µm spheroids, values remained comparable to CTRL, with a transient significant increase in weight and diameter observed at 48 h (*p* < 0.05). Please note that data were analyzed to verify the Gaussian distribution, for this reason the outliers (K > 1.5) were deleted. Student’s *t*-test (two-tailed and heteroscedastic) was used to assess statistical significance between the 6 conditions under testing. *p*-values < *** 0.001.

**Figure 5 biology-14-01381-f005:**
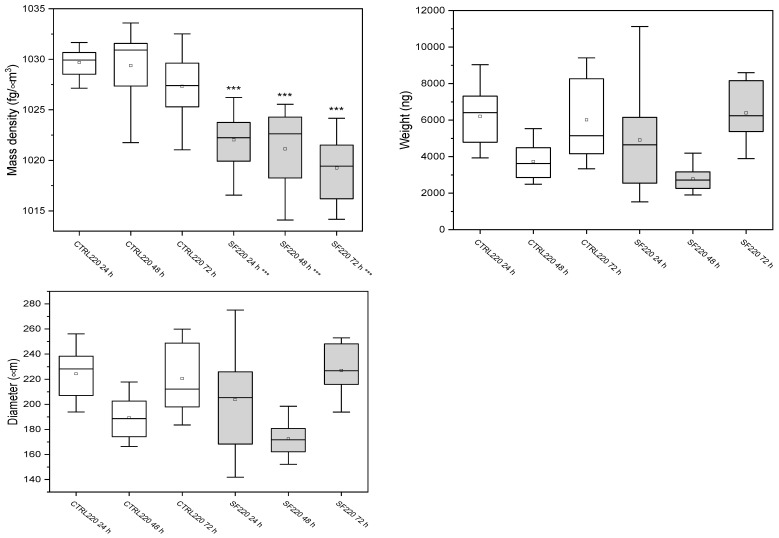
Box plot representation of mass density, weight and diameter of control (CTRL, white bars) and slow-frozen (SF, gray bars) spheroids of 220 µm measured at 24, 48 and 72 h. The line inside each box indicates the median; the lower and upper box boundaries correspond to the first and third quartiles (Q1 and Q3); the whiskers extend to 1.5 × IQR. All parameters are significantly reduced at every time point (*** *p* < 0.001). Data means SD (*n* = 30 per group). Please note that data were analyzed to verify the Gaussian distribution, for this reason the outliers (K > 1.5) are deleted. Student’s *t*-test (two-tailed and heteroscedastic) was used to assess statistical significance between the 6 conditions under testing. *p*-values < *** 0.001.

**Figure 6 biology-14-01381-f006:**
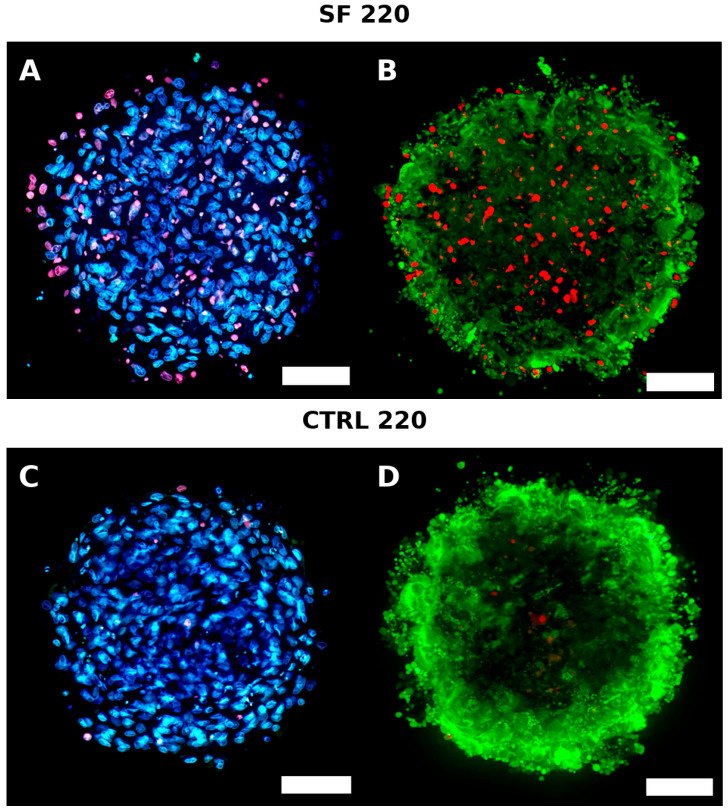
(**A**–**J**) Live/Dead staining 24 h post-thaw. Calcein-AM (green, live cytoplasm), Hoechst 33342 (blue, all nuclei) and propidium iodide (PI, red, dead nuclei) in control (CTRL) and slow-frozen (SF) spheroids. Top row: 140 µm spheroids; bottom row: 220 µm spheroids. CTRL and SF-140 show only scattered PI-positive cells, whereas SF-220 display a dense PI-positive core, indicating central necrosis. Scale bar = 50 µm.

**Figure 7 biology-14-01381-f007:**
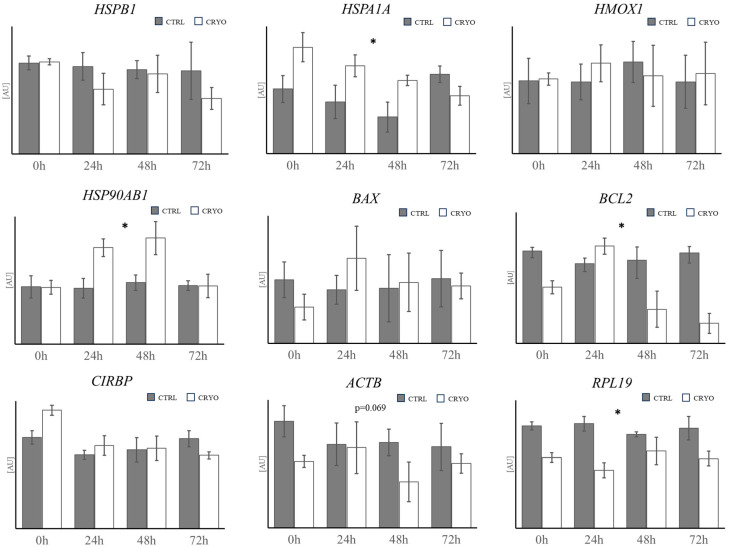
Relative expression of *HSPB1 (HSP27)*, *HSPA1A (HSP70)*, *HMOX1 (HSP32)*, *HSP90AB1*, *BAX*, *BCL2*, *CIRBP*, *ACTB* and *RPL19* in spheroids of 220 µm diameter at 0 h, 24 h, 48 h or 72 h of in vitro culture post-cryopreservation (white column; CRYO) and control spheroids not subjected to cryopreservation (gray column; CTRL) at the same time points. Transcript abundance of target genes was performed after normalization against expression of two reference genes (*YWHAZ* and *SDHA*) and expressed in arbitrary units [AU]. Each column represents the mean value ± SEM of 4 samples (ANOVA General Linear Model * = *p* < 0.05).

**Table 1 biology-14-01381-t001:** Primers used for gene expression analysis by real-time PCR. Bps: base pairs; T: temperature.

Gene	Symbol	Sequence	Accession Number	Annealing T°	Size (bps)
Actin β	*ACTB*	F: 5′ttcctgggtatggatcctg3′ R: 5′ggtgatctccttctgcatcc3′	NM_001009784	60 °C	162
BCL2 associated X Protein	*BAX*	F: 5′ ctccccgagaggtctttttc 3′ R: 5′ tcgaaggaagtccaatgtcc 3′	XM_004015363	58 °C	176
BCL2 apoptosis regulator	*BCL2*	F: 5′ tggatgaccgagtacctgaa3′ R:5′ gccaggagaaatcaaacagg 3′	XM_027960877.2	60 °C	118
Cold inducible RNA binding protein	*CIRBP*	F: 5′ gagggctgagttttgacacc 3′ R: 5′ atgggaagtctgtggatggg 3′	XM_004008776	60 °C	190
Heme oxygenase 1	*HMOX1*	F: 5′ gtcagaggccctgaaggag 3′ R:5′ agggccacgtagatgtggta 3′	XM_027967703.2	60 °C	144
Heat shock protein 90 alpha family class B member 1	*HSP90AB1*	F: 5′ tggagatcaaccctgacca 3′ R: 5′ gggatcctcaagcgagaag 3′	XM_004018854	58 °C	143
Heat shock 70 kDa protein 1A	*HSPA1A*	F: 5′ gttcgacgtgtccatcctga 3′ R: 5′ cagcctgttgtcgaagtcct 3′	NM_001267874	60 °C	100
Heat shock protein family B (small) member 1	*HSPB1*	F: 5′ agctgacggtcaagaccaag 3′ R: 5′ tatttgcgagtgaagcaacg 3′	XM_027961472.2	60 °C	103
Ribosomal protein L9	*RLP19*	F: 5′ caactcccgccagcagat 3′ R:5′ ccgggaatggacagtcaca 3′	XM_004012836	56 °C	127
Succinate dehydrogenase	*SDHA*	F: 5′ catccactacatgacggagca 3′ R: 5′ atcttgccatcttcagttctgcta 3′	XM_012125144	60 °C	90
Tyrosine 3-Monooxygenase	*YWHAZ*	F: 5′ tgtaggatcccgtaggtcatc 3′ R: 5′ ttctctctgtattctcgagcca 3′	NM_001135699	60 °C	168

**Table 2 biology-14-01381-t002:** W8 biophysical read-outs of 140 µm spheroids. Mean ± SD values of mass density (fg µm^−3^), weight (ng) and diameter (µm) for six experimental conditions of 140 µm group: CTRL-24 h, CTRL-48 h, CTRL-72 h and the corresponding slow-frozen spheroids SF-24 h, SF-48 h, SF-72 h. Each value represents ≥30 spheroids measured individually with the W8 Analyzer.

Samples	Mass Density (fg/μm^3^)	Weight (ng)	Diameter (μm)
Ctrl 24 h	1022.6 ± 2.7	1260 ± 385	132 ± 14
Ctrl 48 h	1022 ± 2	1557 ± 613	135 ± 7
Ctrl 72 h	1021.4 ± 2.3	1320 ± 215	140 ± 14
SF 24 h	1022 ± 2	1666 ± 260	140 ± 20
SF 48 h	1022 ± 2	1508 ± 428	146 ± 8
SF 72 h	1022 ± 2	1394 ± 579	135 ± 19

**Table 3 biology-14-01381-t003:** W8 biophysical read-outs of 220 µm spheroids. Mean ± SD values of mass density (fg µm^−3^), weight (ng) and diameter (µm) for six experimental conditions of 220 µm group: CTRL-24 h, CTRL-48 h, CTRL-72 h and the corresponding slow-frozen spheroids SF-24 h, SF-48 h, SF-72 h. Each value represents ≥ 30 spheroids measured individually with the W8 Analyzer.

Samples	Mass Density (fg/μm^3^)	Weight (ng)	Diameter (μm)
Ctrl 24 h	1029.7± 1.3	6207 ± 1501	224 ± 19
Ctrl 48 h	1029 ± 4	3735 ± 928	189 ± 15
Ctrl 72 h	1027.3 ± 3.4	6019 ± 21,170	221 ± 27
SF 24 h	1022.0 ± 2.6	4014 ± 2547	204 ± 35
SF 48 h	1020.5 ± 3.8	3273 ± 1310	180 ± 22
SF 72 h	1019.3 ± 3.3	6397 ± 1719	227 ± 21

## Data Availability

The data produced during the current study are available from the corresponding author on reasonable request.

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
