# Peer review of "Evaluating the Response to Cryopreservation of Ovine Fibroblast Spheroids"

_biology, 2025, doi:10.3390/biology14101381_

Round 1

Reviewer 1 Report (Previous Reviewer 2)

Comments and Suggestions for Authors

1. Authors are asked to add more detailed methodology for Spheroid Generation especially specific instrumentation used for generating Spheroid and point out the rpm used. 

2. Authors examined the cryopreservation of spheroids maximum upto 7 days only. Did authors checked cryopreserving spheroids in prolonged time period notably 3 to 12 months. 

3. Formatting is not good. Authors are asked to follow the journal format strictly. It is very difficult to read the manuscript. Table 3 all the reading are overlapped and very difficult to read. 

4. Authors clearly explain the Table 3, why increasing diameter of the spheroid not influence the mass density. What it indicates? Is there any correlation between them.

5. Why statistical analysis appeared two times in materials and methods - (Page 8 - line 240-244 and Page 11 - line 292 - 298)

6. In Alamar Blue® Assay (line 171), 'R' should be removed. 

7. There are lot of grammatical and typo errors throughout the manuscript (Page 29 - line 572-573). Authors are asked to correct and revise the manuscript.  

Author Response

We thank all Reviewers for their careful evaluation of our manuscript and for the constructive comments. We have revised the manuscript accordingly. Below we provide a point-by-point response to each comment. Reviewer comments are reported in italics, while our responses are provided below each point.

Reviewer 1

  1. Authors are asked to add more detailed methodology for Spheroid Generation especially specific instrumentation used for generating Spheroid and point out the rpm used.

Response: Line 144
Prior to cell seeding, Corning® Elplasia® 96-well round-bottom microcavity plates (400 microcavities per well) were pre-wetted to eliminate residual air bubbles within the microwells. Each well was filled with 50 µL of complete growth medium (DMEM supplemented with 10% FBS), followed by centrifugation at 500 × g for 1 min at room temperature. This pre-wetting ensured uniform filling of the microcavities and promoted consistent distribution of cells during subsequent seeding.

  1. Authors examined the cryopreservation of spheroids maximum up to 7 days only. Did authors check cryopreserving spheroids in prolonged time period notably 3 to 12 months?

Response:
In our study, we experimentally evaluated cryostorage of spheroids up to 7 days, which was sufficient to demonstrate recovery and viability post-thaw under the tested conditions. We did not extend the experimental window to several months; however, once transferred into liquid nitrogen, cryogenic conditions remain stable over time. This principle is well established in cryobiology, where samples such as mammalian embryos have been successfully stored and recovered even after 20 years of cryopreservation. Therefore, while our study was limited to short-term evaluation, there is strong evidence that spheroids preserved under liquid nitrogen would remain stable for prolonged periods (3–12 months or longer), provided that storage conditions are continuously maintained.

  1. Formatting is not good. Authors are asked to follow the journal format strictly. It is very difficult to read the manuscript. Table 3 readings are overlapped and very difficult to read.

Response:
The manuscript has been reformatted according to the journal guidelines. Table 3 has been corrected to ensure clarity and readability.

  1. Authors clearly explain Table 3, why increasing diameter of the spheroid did not influence the mass density. What does it indicate? Is there any correlation between them?

Response:
Table 3 reports the mean values of spheroid mass density, weight, and diameter for the 220 µm category, which includes both cryopreserved spheroids and their respective controls at 24, 48, and 72 h. Within this group, no statistically significant variation was observed in the spheroid diameter, indicating that an increase in diameter did not influence the measured mass density. The only statistically significant differences detected were in the mass density values between cryopreserved spheroids and controls across all time points.

  1. Why does statistical analysis appear two times in Materials and Methods? (Page 8, line 240-244 and Page 11, line 292-298).

Response:
We thank the Reviewer for the suggestion. A dedicated paragraph describing the statistical analysis has been added as a single section in Materials and Methods.

  1. In Alamar Blue® Assay (line 171), “R” should be removed.

Response:
Thank you. All redundant ® symbols have been removed.

  1. There are many grammatical and typographical errors throughout the manuscript (e.g., Page 29, line 572-573). Authors are asked to correct and revise the manuscript.

Response:
The manuscript has been thoroughly revised and all grammatical and typographical errors have been corrected.

Reviewer 2 Report (Previous Reviewer 1)

Comments and Suggestions for Authors

Dear authors

I understand this is a new submission, but the file you sent appears to be a revision with track changes. I believe a new submission should be in the standard format, not the one you sent.

Here are some discussion of the manuscript as described:

The study focuses exclusively on ovine (sheep) fibroblasts. While this is a valid model, the findings may not be generalizable to other cell types (e.g., stem cells, hepatocytes) or other species, which are also crucial in tissue engineering.

The research compares just two spheroid sizes (140 µm and 220 µm). A more comprehensive study would investigate a wider range of sizes to better understand the relationship between spheroid size and cryopreservation success. The chosen sizes might not be representative of the full range used in the field.

The abstract states that the 220 µm spheroids showed a decline in vitality and mass density, but it doesn't offer a clear explanation of *why*. For instance, is it due to poor cryoprotectant penetration, formation of larger ice crystals in the core, or something else? The study could be strengthened by exploring the underlying mechanisms.

The abstract lists a wide range of parameters but doesn't detail the specific findings for each one beyond the general conclusion. For example, it mentions gene expression changes but doesn't state which genes or pathways were affected, which is a significant piece of information.

With only two size groups, the study might lack the statistical power to draw robust conclusions, especially if the sample sizes within each group are small. The abstract doesn't mention the number of replicates or independent experiments.

The study looks at three time points "after thawing." The abstract doesn't specify how long after thawing these points were, which is important for understanding long-term viability and recovery. A true measure of success would include monitoring the spheroids for weeks or even months to see if they can fully recover and function.

Author Response

Reviewer 2

  1. The study focuses exclusively on ovine fibroblasts. While this is a valid model, the findings may not be generalizable to other cell types or other species.

Response:
We thank the Reviewer for this important observation. Our study was intentionally focused on ovine fibroblast spheroids as a first model because fibroblasts are abundant, easy to isolate, and play a central role in connective tissue biology and wound healing. Importantly, fibroblasts are also characterized by strong intercellular junctions and abundant extracellular matrix production, which makes them relatively resistant to mechanical and osmotic stress but, at the same time, more challenging in terms of cryoprotectant penetration. For this reason, they represent a stringent model for assessing the impact of cryopreservation on 3D multicellular structures. While we recognize that our findings may not be directly generalizable to other cell types or species, we believe that demonstrating cryopreservation feasibility in fibroblast spheroids provides a solid foundation for extending these approaches to more delicate cell types, such as stem cells or hepatocytes, in future studies.

  1. The research compares just two spheroid sizes (140 µm and 220 µm). A more comprehensive study would investigate a wider range of sizes.

Response:
We thank the Reviewer for the insightful comment. We agree that a broader panel of spheroid sizes would provide a more detailed overview of the relationship between spheroid diameter and cryopreservation outcome. In this study, however, we deliberately selected two distinct size groups (≈140 µm and ≈220 µm) that lie around the diffusion threshold reported for avascular tissues (~200 µm). By bracketing this critical boundary, we aimed to generate proof-of-concept evidence of a size-dependent effect without introducing excessive experimental variability.

  1. The Abstract states that the 220 µm spheroids showed a decline in vitality and mass density, but it does not offer a clear explanation of why.

Response:
We thank the Reviewer for the helpful suggestion. A sentence has been added to the Abstract to clarify the possible reasons for the decline in vitality and mass density observed in 220 µm spheroids, namely the limited penetration of cryoprotectants and core-related stress.

  1. The Abstract lists a wide range of parameters but does not detail the specific findings for each one.

Response:
We thank the Reviewer for this useful comment. We have revised the Abstract to briefly summarize the key findings for each parameter. In particular, we now indicate that cryopreservation upregulated stress-related genes such as HSPA1A (HSP70) and HSP90AB1, while downregulating the anti-apoptotic gene BCL2.

  1. With only two size groups, the study might lack the statistical power to draw robust conclusions. The Abstract does not mention the number of replicates or independent experiments.

Response:
We thank the Reviewer for the observation. While we agree that the number of replicates is important for assessing robustness, it is generally considered methodological detail and therefore not included in the Abstract. Instead, this information is clearly reported in the Materials and Methods.

  1. The study looks at three time points “after thawing.” The Abstract does not specify how long after thawing these points were.

Response:
We thank the Reviewer for the comment. The exact timing of the post-thaw measurements (24 h, 48 h, and 72 h) is reported in the Materials and Methods and in the figure legends. Since the Abstract is intended to provide only a concise overview of the study and its main findings, we preferred to keep detailed timing information in the methodological section.

Reviewer 3 Report (New Reviewer)

Comments and Suggestions for Authors

The manuscript “Evaluating the response to cryopreservation of Ovine Fibro-2 blast spheroids” by Piras et al. investigates the effects of cryopreservation via slow freezing on spheroids of two sizes (140 and 220 μm). Viability, adhesion, spheroid mass density, gene expression, and several other parameters were analyzed at three time points after thawing and compared with control spheroids. Overall, this is an interesting and useful study, but the discussion and comparison with previous works is largely missing.

Major Comments:

  1. Note to the authors, or possibly to the editors: the provided PDF is very messy due to the presentation of deleted text/images, which sometimes overlap with the new text (see page 13, for example). It is highly inconvenient to review documents in this form.
  2. There are previous works addressing cryopreservation of different types of spheroids (10.1371/journal.pone.0230428, 10.1007/s11814-022-1279-9, 10.1021/acsami.2c18288, 10.1038/s41598-022-23913-3). The authors should provide a clear description of the key findings of these works and compare their results with them in the discussion.
  3. The standard deviation of spheroid size (for the 140 and 220 μm groups) should be indicated somewhere.
  4. Section 3.4 (Live/dead imaging) would benefit from quantitative analysis, namely the % of dead cells in the center vs. periphery of the spheroid (e.g., separated by 50% of the spheroid radius).
  5. Line 612: “upregulation of RPL19 in control spheroids” – this is rather downregulation in the thawed spheroids.
  6. Abbreviation list: contains incorrect items.
  7. Section 3.1 title: “Repleting” – typo.
  8. Figure 5 caption: “Please note that were” – stylistic error. I also think the deleted versions of Figures 4 and 5 (with results grouped by control vs. treatment) were clearer than the current versions, but I leave it to the authors to decide.
  9. Figure 2 is inconsistent: panels A, B, C, D are labeled, but C and D also include titles (CTR…), while A and B do not.

Author Response

We thank all Reviewers for their careful evaluation of our manuscript and for the constructive comments. We have revised the manuscript accordingly. Below we provide a point-by-point response to each comment. Reviewer comments are reported in italics, while our responses are provided below each point.

Reviewer 3

  1. The manuscript is messy due to track changes and overlapping deleted/added text.

Response:
We apologize for the inconvenience. A clean, revised version of the manuscript has now been prepared and submitted.

  1. There are previous works addressing cryopreservation of different types of spheroids (references provided). Authors should describe the key findings of these works and compare their results with the present study.

Response: line 530
We thank the Reviewer for the suggestion. A dedicated paragraph has been added to the Discussion, summarizing the key findings of the cited works and comparing them with our results.

  1. The standard deviation of spheroid size (for the 140 and 220 µm groups) should be indicated.

Response:
We thank the Reviewer for the comment. The standard deviation of spheroid diameter is now reported in Tables 2 and 3.

  1. Section 3.4 (Live/Dead imaging) would benefit from quantitative analysis.

Response:
We thank the Reviewer for this valuable suggestion. We agree that a quantitative analysis of the spatial distribution of dead cells would provide additional insight. However, this type of measurement was not feasible in the present study because Live/Dead staining was intended as a qualitative assay and the setup did not include image segmentation tools. Instead, we used complementary approaches (Alamar Blue assay and W8 biophysical profiling) to quantify viability and structural integrity. We acknowledge this as a limitation and consider quantitative imaging an important improvement for future work.

  1. Line 612: “upregulation of RPL19 in control spheroids” – this is rather downregulation in the thawed spheroids.

Response:
We thank the Reviewer for pointing this out. The errors have been corrected.

  1. Abbreviation list contains incorrect items.

Response:
We thank the Reviewer for pointing this out. The list of abbreviations has been corrected.

  1. Section 3.1 title: “Repleting” – typo.

Response:
We thank the Reviewer for pointing this out. The typo has been corrected to “Replating.”

  1. Figure 5 caption: “Please note that were” – stylistic error. Also, the previous versions of Figures 4 and 5 may have been clearer.

Response:
We thank the Reviewer for the observation. The stylistic error in the caption of Figure 5 has been corrected. Regarding the layout, we decided to maintain the current presentation, as it allows direct comparison across time points, while ensuring clarity in the legend and supporting tables.

  1. Figure 2 is inconsistent: panels A, B, C, D are labeled, but C and D also include titles (CTRL…), while A and B do not.

Response:
We thank the Reviewer for the comment. Panels A and B of Figure 2 are labeled with “SF” (slow frozen) to indicate the treated groups, while panels C and D are labeled with “CTRL” for the control groups. This explains the apparent difference in panel titles.

Round 2

Reviewer 1 Report (Previous Reviewer 2)

Comments and Suggestions for Authors

Authors addressed all the queries and now it is fine for publication.

Author Response

Thanks

Reviewer 3 Report (New Reviewer)

Comments and Suggestions for Authors

Fig. 5. What does "Data means SD" mean? This is a box plot where median and quartiles are indicated? Please clarify (in Fig. 4 too)

Author Response

Comment 1:Fig. 5. What does "Data means SD" mean? This is a box plot where median and quartiles are indicated? Please clarify (in Fig. 4 too)

Response 1: 

We thank the Reviewer for pointing out this inconsistency. Figures 4 and 5 represent box plots, in which the line inside the box indicates the median, the box limits correspond to the first and third quartiles, and the whiskers extend to 1.5 × IQR, Therefore, the label “Data means ± SD” was misleading and has been corrected in the figure legends of both Figure 4 and Figure 5 to accurately describe the representation of the data.

This manuscript is a resubmission of an earlier submission. The following is a list of the peer review reports and author responses from that submission.

Round 1

Reviewer 1 Report

Comments and Suggestions for Authors

This manuscript presents a well-structured study that investigates the critical impact of cryopreservation on ovine fibroblast spheroids. This topic is highly relevant to tissue engineering and regenerative medicine, as the research effectively addresses a key challenge in this field: maintaining the integrity and functionality of 3D cell structures after freezing and thawing.

I have several questions regarding this work:

  1. Could the spheroid formation method itself influence the size and internal structure, and consequently, their response to cryopreservation?

  2. Why were 140 µm and 220 µm specifically chosen as the spheroid sizes for investigation? Are these sizes representative of typical applications, or were they selected to highlight a particular hypothesis about size effects?

  3. Regarding the replanting assay, how long were cells allowed to adhere and proliferate, and how was this quantified?

  4. Which specific genes were analyzed by RT-PCR? Were these genes related to stress response, viability, extracellular matrix production, or specific differentiation markers?

  5. Could you elaborate on the principles behind the W8 Biophysical Analyzer for quantifying spheroid mass density, weight, and diameter? How does it ensure accuracy, especially for such delicate spheroid structures?

  6. Were unfrozen spheroids of both sizes used as controls for all assays to establish baseline parameters?

  7. For the 220 µm spheroids, what was the magnitude of the decline in vitality and mass density compared to the 140 µm spheroids or unfrozen controls?

  8. Beyond "mass density" and "vitality," what other specific biophysical parameters were affected, particularly in the 220 µm spheroids?

  9. Do the authors anticipate these findings regarding spheroid size to be generalizable to spheroids derived from other cell types relevant to regenerative medicine?

Author Response

Please see the attachment for a more detailed reply.

Reviewer 2 Report

Comments and Suggestions for Authors

1. Authors are carefully write the results neatly and sequentially. Simply arranging figures with legend is not acceptable. 

2. Authors are asked to expand the abbreviations during the first introduction of the corresponding abbreviations. CTRL and SF need to be expanded for better understanding. 

3. Authors are asked to strictly follow the journal format. 

4. There is no figure legend for Figure 5.

5. Scale bar is missing in Figure 7.

6. Why both Calcein-AM and Hoechst used for Live dead staining?

7. There is no X-axis and Y-axis title in the graph. There is no figure number and legend for the last figure. 

8. Why qPCR is preferred and not western blotting techniques to confirm the results?

9. It is very difficult to read the discussion part. References are not cited as per the journal format. 

10. Authors are asked to make all this corrections and resubmit for review.

Comments on the Quality of English Language

Writing style is Poor. Authors not followed the journal guidelines for preparing this manuscript.

1. Authors are carefully write the results neatly and sequentially. Simply arranging figures with legend is not acceptable. 

2. Authors are asked to expand the abbreviations during the first introduction of the corresponding abbreviations. CTRL and SF need to be expanded for better understanding. 

3. Authors are asked to strictly follow the journal format. 

4. There is no figure legend for Figure 5.

5. Scale bar is missing in Figure 7.

6. Why both Calcein-AM and Hoechst used for Live dead staining?

7. There is no X-axis and Y-axis title in the graph. There is no figure number and legend for the last figure. 

8. Why qPCR is preferred and not western blotting techniques to confirm the results?

9. It is very difficult to read the discussion part. References are not cited as per the journal format. 

10. Authors are asked to make all this corrections and resubmit for review.

Author Response

(The authors gave the same response as above.)

Reviewer 3 Report

Comments and Suggestions for Authors

Piras et al. provide a manuscript about their work concerning cryopreservation of spheroids from ovine fibroblasts. They cryopreserved spheroids of two different sizes and following a number of tests, they find that the larger spheroids are more susceptible to cryopreservation-induced damage. It is very positive that the authors did a number of tests on the spheroids to characterize the effects of the cryopreservation. Unfortunately, the methods and results are not sufficiently well described to judge the validity of the results and their interpretation.

The format of the results section and part of the discussion does not meet scientific standards. Figures are lacking captions that describe the results presented in there.

The results should contain all description of data and can contain some interpretation.

The discussion should contain a discussion of the results in a wider context. No results should be mentioned in the discussion that were not described in the results section.

Authors should also carefully go through the text and figures a number of overseights, such as grammatical and spelling errors or formatting of references. Authors also randomly use , and . as decimal separators.

Specific comments:

Line 119: How were cells harvested? There would be the danger of damaging the spheroid during this process. Why is there no control for this process?

Lines 127-128: It is described that the spheroids are placed directly in -80°C, whereas the graphical abstract seems to indicate that they were first kept at -20°C

Lines 125-132: The cryopreservation process (cooling and storage) took about 48h. What happened to control cells during this time? Were they allowed to further grow? This would make an uneven comparison for some of the test, such as size, weight, metabolic activity.

Line 140: How were controls treated? Have they been growing for 48h longer then cryopreserved spheroids?

Line 146: Why has the live/dead assay only been performed after 24h? Some dead cells might be expelled from the spheroids at this time point, as the authors speculate themselves in the discussion.

Lines 150-151: Details of the microscopy are missing. At least objective and fluorescence channels should be described.

Lines 153-156: The basis of this measurement needs to be described. Size measurements seem to be inconsistent with the microscopy images provided.  E.g. the spheroid in figure 2B seems to be 300-400 um in diameter, 2C > 200 um, 7G,H,I,L > 200 um. All of these sizes are far outside of the range given in the measurements provided in figure 4 and 5.

It should also be justified somewhere at the beginning on which basis the spheroids are called 140- and 220-um sized.

Figure 2A, C: The cryopreserved spheroids look much larger than the controls. Have they been cultivated longer?

Figure 2D: It should be shown, how a spheroid looks like that does not form a monolayer.

Figure 4: What is the authors explanation that the spheroids 48h after cryopreservation are highly significantly larger and heavier than the controls (assuming that the grey bars represent cryopreserved samples)? Error bars are much smaller vs the other time points. This might point at some technical difficulties for the other time points?

Line 248: The whole data should be presented, given the seemingly inconsistent results. At least it should be mentioned, where and how many outliers have been removed.

Lines 251-253: Why has Students t-test been used here? The methods section mentions ANOVA, which seems more appropriate to compare the 6 conditions.

Figure7 L: This spheroid looks much more yellow than the others. Why is this? Is there a stronger PI staining. Individual fluorescence channels should be shown, since merged images are often difficult to interpret.

Lines 310-317: The number of dead cells need to be quantified, if authors which to draw conclusions on this.

Lines 312-314: In the example dead cells seem to be not only in the core. This needs to also be quantified, if authors want to conclude this.

Figure(s) after Figure 7: It is not clear, which spheroids are tested here. Why is there no comparison between the two different sizes?

Line 354: The normalization procedure needs some clarification. How was this exactly done? The methods section mentions normalization against 4 genes, one of them (RPL19) is shown to change. Was the normalization still performed using this gene?

Author Response

(The authors gave the same response as above.)
